# Occult Vertebral Fracture (OVF) in Patients Who Underwent Hepatectomy for Colorectal Liver Metastasis: Strong Association with Oncological Outcomes

**DOI:** 10.3390/cancers15235513

**Published:** 2023-11-22

**Authors:** Kenei Furukawa, Koichiro Haruki, Tomohiko Taniai, Mitsuru Yanagaki, Masashi Tsunematsu, Yoshiaki Tanji, Shunta Ishizaki, Yoshihiro Shirai, Shinji Onda, Toru Ikegami

**Affiliations:** Division of Hepatobiliary and Pancreas Surgery, Department of Surgery, The Jikei University School of Medicine, 3-25-8, Nishi-Shinbashi, Minato-ku, Tokyo 105-8461, Japan; haruki@jikei.ac.jp (K.H.); ahiru_tt_1009@yahoo.co.jp (T.T.); m.yana0109@gmail.com (M.Y.); tsunematsu@jikei.ac.jp (M.T.); yoshiaki.leo.0819@gmail.com (Y.T.); s.ithsju@gmail.com (S.I.); shirai@jikei.ac.jp (Y.S.); s-onda@jikei.ac.jp (S.O.); toruikegamijikei@jikei.ac.jp (T.I.)

**Keywords:** occult vertebral fracture, OVF, sarcopenia, osteopenia, osteosarcopenia, colorectal liver metastases, hepatectomy

## Abstract

**Simple Summary:**

The impact of preoperative occult vertebral fracture (OVF) on oncological outcomes has not been investigated. We investigated the prognostic significance of OVF on the long-term outcomes of patients with colorectal liver metastases (CRLM) after hepatectomy. OVF was evaluated using preoperative computed tomography. OVF was diagnosed in 48 of 140 patients (34%). Multivariate analysis demonstrated that OVF was an independent factor for both disease-free and overall survival. A higher age, adjuvant chemotherapy for a primary lesion before metachronous liver metastases, osteopenia, and hypoalbuminemia were independent risk factors for OVF. The preoperative evaluation of OVF can be a useful prognostic marker for risk stratification and clinical decision-making in patients who underwent hepatectomy for CRLM.

**Abstract:**

Aim: The impact of occult vertebral fracture (OVF) on oncological outcomes after surgery has not been investigated, although its significance in orthopedics has been much debated recently. We evaluated the prognostic significance of OVF on the long-term outcomes of patients with colorectal liver metastases (CRLM) after hepatectomy. Methods: We included 140 patients with CRLM who underwent hepatectomy. OVF was identified using quantitative measurement and preoperative sagittal computed tomography image reconstruction from the 11th thoracic vertebra to the 5th lumber vertebra. Results: OVF was identified in 48 (34%) of the patients. In multivariate analysis, lymph node metastases (*p* < 0.01), multiple tumors (*p* = 0.02), extrahepatic lesions (*p* < 0.01), OVF (*p* < 0.01), intraoperative bleeding (*p* = 0.04), and curability 1 or 2 (*p* < 0.01) were independent and significant predictors of disease-free survival and extrahepatic lesions (*p* < 0.01), osteosarcopenia (*p* = 0.02), and OVF (*p* < 0.01) were independent and significant predictors of overall survival. A higher age, adjuvant chemotherapy for a primary lesion before metachronous liver metastases, osteopenia, and hypoalbuminemia were independent risk factors for OVF. Conclusions: The evaluation of preoperative OVF is a useful prognostic factor for risk stratification and clinical decision-making for patients with CRLM.

## 1. Introduction

Colorectal cancer is the second leading cause of cancer-related deaths worldwide and the third most common cancer [1]. Forty to fifty percent of patients will develop colorectal liver metastases (CRLM) during their course [2]. Nevertheless, hepatic resection improves the survival expectancy compared with systemic therapies, and their 10-year survival rate has reached 15–25% [3,4].

Reliable prognostic indicators after hepatectomy for CRLM are needed for improving prognosis, as the outcomes are unfavorable and unsatisfactory. Tumor factors such as the tumor number, size, tumor maker, lymph node metastases, extrahepatic metastatic disease, and genomic information are significant factors [5,6]. In addition, patient-related factor, such as sarcopenia characterized by loss of skeletal muscle, can have a significant impact on the prognosis of malignancies including hepatocellular carcinoma and CRLM [7,8,9].

Sarcopenia has been linked to low bone mineral density (BMD), known as osteopenia, which is another important patient-related factor in cancer treatment. We previously demonstrated that osteopenia or osteosarcopenia, the concomitant occurrence of sarcopenia and osteopenia, was an independent risk factor for poor prognosis in patients with esophageal, colorectal, pancreatic, and hepatic cancers, including CRLM [10]. Osteopenia or osteoporosis is associated with a higher risk of fractures, and vertebral fractures are the most common type [11]. However, vertebral fractures are often missed and two-thirds of them are not clinically detected [12]. Moreover, the impact of vertebral fractures on the prognosis for malignancies has not been reported yet.

The aim of this study is to define a vertebral fracture diagnosed using preoperative computed tomography (CT) screening as an occult vertebral fracture (OVF) and investigate the prognostic impact of OVF along with other prognostic factors, including osteopenia, sarcopenia, and osteosarcopenia, on the oncological outcomes in patients with CRLM after hepatectomy.

## 2. Methods

### 2.1. Patients

The subjects were 140 patients without unresectable extrahepatic tumors who underwent hepatic resection for what could be defined as resectable CRLM after preoperative chemotherapy at the Department of Hepatobiliary and Pancreas Surgery, the Jikei University Hospital, Tokyo, Japan between May 2007 and March 2019. Patients with symptomatic vertebral fracture, other bone diseases, and a lack of data were excluded from analysis. Parenchymal-sparing hepatectomy was generally performed to preserve an estimated residual liver volume of more than 30%. If the estimated residual liver volume was less than 30%, percutaneous transhepatic portal vein embolization was performed. Unresectable CRLM is defined as technically difficult to resect and borderline resectable CRLM is defined as CRLM with extrahepatic tumors or H2 or H3 liver metastases [13]. In general, preoperative chemotherapy was performed for unresectable and borderline resectable CRLM. Postoperative chemotherapy was administered depending on the patient’s general condition and the attending physician’s discretion. Further surgical procedures, treatment for recurrence, the definition of postoperative complications, and postoperative surveillance were described in our previous report [10].

This retrospective study using a prospectively maintained database was approved by the Human Ethics Committee of the Jikei University School of Medicine. Informed consent from patients was waived because this study was retrospective, using anonymized data.

### 2.2. Definition of Osteopenia, Sarcopenia, Osteosarcopneia, and Occult Vertebral Fracture

Osteopenia was defined using bone mineral density (BMD) according to the previous report [10]. Sarcopenia was defined using psoas muscle mass area (PMA) according to the previous report [14]. Osteosarcopenia was defined as the concomitant occurrence of osteopenia and sarcopenia. Occult vertebral fracture (OVF) was identified using quantitative measurement [15] and preoperative sagittal CT image reconstruction. The anterior (A), central (C), and posterior (P) heights were calculated (vertebrae from the 11th thoracic vertebra to the 5th lumber vertebra). The criteria for OVF were C/A < 0.8 or C/P < 0.8 regardless of fracture history (Figure 1).

### 2.3. Systemic Inflammatory Biomarkers

We preoperatively assessed the Glasgow Prognostic Score (GPS) [16] and prognostic nutrition index (PNI) [17] as other systemic inflammatory biomarkers. The GPS included the serum albumin (Alb) and serum C-reactive protein (CRP). The GPS was determined as follows: GPS of 0, Alb ≥ 3.5 g/dL and CRP ≤ 1.0 mg/dL; GPS of 1, Alb < 3.5 g/dL or CRP > 1.0 mg/dL; GPS of 2, Alb < 3.5 g/dL and CRP > 1.0 mg/dL. The PNI was calculated as “serum Alb level (g/L) + 0.005 × total lymphocyte count”.

### 2.4. Risk Factors Associated with Survival and OVF

For analyses for risk factors associated with survival and OVF, the clinicopathological data included age, gender, body mass index, lymph node metastases, timing of CRLM, resectability, neoadjuvant chemotherapy, tumor number, tumor size, extrahepatic lesions, osteopenia, sarcopenia, osteosarcopenia, OVF, GPS, PNI, serum CEA level, serum Alb level, operative time, intraoperative bleeding, infectious postoperative complications, postoperative stay, curability, and treatment of recurrence. Based on the receiver operating characteristic curve coordinates, the most optimal cut-off points for continuous variables were determined.

### 2.5. Statistical Analysis

Data were expressed as medians with interquartile ranges. The Mann–Whitney U test was used to compare continuous variables, whereas categorical variables were analyzed using the chi-square test. Long-term outcomes including DFS, OS, and cancer-specific survival were estimated using the Kaplan–Meier method with the log-rank test. The risk factors for survival were analyzed using the Cox proportional hazards regression models. The risk factors for OVF were analyzed using multivariate logistic regression models. Two-sided *p*-values less than 0.05 were considered statistically significance. These analyses were performed using IBM^®^ SPSS Statistics version 25.0 (IBM Japan, Tokyo, Japan).

## 3. Results

### 3.1. Patients’ Characteristics

This study included 140 patients (96 men and 44 women). The median age was 66 (59–73) years. Resectable, borderline resectable, and unresectable CRLM were initially diagnosed in 76 patients (54%), 61 patients (44%), and 3 patients (2%), respectively. The number of patients with preoperative chemotherapy was 48 (34%) and that of patients with adjuvant chemotherapy after hepatectomy was 27 (19%). Osteopenia, sarcopenia, and osteosarcopenia were diagnosed in 77 patients (55%), 69 patients (49%), and 43 patients (31%), respectively. OVF was diagnosed in 48 patients (34%) and the median ratios of C/A and C/P were 0.72 with a range of 0.44 to 1.29 and 0.76 with a range of 0.48 to 1.0, respectively, in patients with OVF. The bone status included healthy bone in 48 patients (34%), osteopenia without OVF in 44 patients (31%), non-osteopenic OVF in 15 patients (11%), and osteopenic OVF in 33 patients (24%).

### 3.2. Clinicopathological Variables Associated with DFS after Hepatectomy for CRLM Using Univariate and Multivariate Analyses

Table 1 summarizes the clinicopathological variables associated with DFS after hepatic resection for CRLM. The univariate analysis showed that DFS was significantly worse in patients with lymph node metastases (*p* < 0.01), multiple tumors (*p* < 0.01), extrahepatic lesions (*p* < 0.01), OVF (*p* < 0.01), a lot of intraoperative bleeding (*p* = 0.047), and R1 or 2 (*p* < 0.01). In the multivariate analysis, lymph node metastases (hazard ratio, 1.89; 95% confidence interval, 1.18–3.01; *p* < 0.01), multiple tumors (hazard ratio, 1.28; 95% confidence interval, 1.03–1.57; *p* = 0.02), extrahepatic lesions (hazard ratio, 1.74; 95% confidence interval, 1.34–2.26; *p* < 0.01), OVF (hazard ratio, 1.70; 95% confidence interval, 1.38–2.11; *p* < 0.01), a lot of intraoperative bleeding (hazard ratio, 1.25; 95% confidence interval, 1.02–1.55; *p* = 0.04), and R1 or 2 (hazard ratio, 1.51; 95% confidence interval, 1.16–1.96; *p* < 0.01) were independent and significant predictors of DFS.

### 3.3. Clinicopathological Variables Associated with OS after Hepatectomy for CRLM Using Univariate and Multivariate Analyses

Table 2 summarizes the clinicopathological variables associated with OS after hepatic resection for CRLM. The univariate analysis showed that OS was significantly worse in patients with lymph node metastases (*p* = 0.04), multiple tumors (*p* = 0.03), extrahepatic lesion (*p* < 0.01), sarcopenia (*p* = 0.03), osteosarcopenia (*p* < 0.01), OVF (*p* < 0.01), GPS 1 or 2 (*p* = 0.03), a low PNI (*p* = 0.03), a long operative time (*p* < 0.01), and infectious postoperative complications (*p* = 0.01). In the multivariate analysis, extrahepatic lesions (hazard ratio, 1.79; 95% confidence interval, 1.28–2.51; *p* < 0.01), osteosarcopenia (hazard ratio, 2.54; 95% confidence interval, 1.15–5.57; *p* = 0.02), and OVF (hazard ratio, 1.92; 95% confidence interval, 1.43–2.58; *p* < 0.01) were independent and significant predictors of OS.

### 3.4. Impact of Osteopenia, Osteosarcopenia, and OVF on DFS and OS after Hepatectomy for CRLM

The DFS of patients with OVF was significantly lower than that of patients without OVF (*p* < 0.01) (Figure 2E). The OS of patients with osteosarcopenia was significantly lower than that of patients without osteosarcopenia (*p* < 0.01; 3-year and 5-year survival, 60.6% vs. 84.0% and 44.8% vs. 66.4%, respectively) (Figure 2D). The OS of patients with OVF was significantly lower than that of patients without OVF (*p* < 0.01; 3-year and 5-year survival, 47.8% vs. 86.1% and 30.3% vs. 69.1%, respectively) (Figure 2F). There were no significant differences in survival (Figure 2A–C).

In terms of the impact of bone status on DFS and OS, the DFS and OS of patients with OVF was significantly lower than those of patients without OVF regardless of osteopenia (*p* < 0.01). The OS was comparable between patients with healthy bone and osteopenia without OVF (*p* = 0.65; 3-year and 5-year survival, 87.2% vs. 85.0% and 76.8% vs. 70.0%, respectively) (Figure 3). The rate of 5-year mortality after hepatectomy was as follows: no osteopenia and no OVF (34.2%, n = 38), osteopenia and no OVF (41.9%, n = 31), no osteopenia and OVF (80.0%, n = 10), osteopenia and OVF (87.0%, n = 23) (Figure 4).

### 3.5. Association between Clinical Variables and OVF and Risk Factors for OVF Using Multivariate Logistic Regression Analysis

Table 3 shows the association between clinical variables and OVF. Patients with OVF were significantly older and more commonly had adjuvant chemotherapy for a primary lesion before metachronous liver metastases, osteopenia, and a lower Alb than those without OVF (*p* = 0.02, 0.01, 0.02, and 0.01, respectively). There were no significant differences in osteosarcopenia and treatment of recurrence between the two groups.

Multivariate analysis showed that old age (odds ratio 3.02, 95% confidence interval 1.35–6.75, *p* < 0.01), adjuvant chemotherapy for a primary lesion before metachronous liver metastases (odds ratio 3.03, 95% confidence interval 1.20–7.67, *p* = 0.02), osteopenia (odds ratio 2.71, 95% confidence interval 1.19–6.15, *p* = 0.02), and low serum Alb level (odds ratio 2.34, 95% confidence interval 1.04–5.28, *p* = 0.04) were significant independent predictors of OVF (Figure 5).

## 4. Discussion

This study demonstrated that the disease-free and overall survival following hepatic resection for CRLM were significantly shorter in patients with OVF than in those without OVF. Multivariate analysis further showed that OVF was an independent factor for disease-free and overall survival. Moreover, a higher age, adjuvant chemotherapy for a primary lesion before metachronous liver metastases, osteopenia, and hypoalbuminemia were independent risk factors for OVF. These findings suggested that preoperative OVF was associated with long-term outcomes in patients who underwent hepatic resection for CRLM, and concomitant OVF should be considered in patients over age 70, with a history of chemotherapy, osteopenia, and malnutrition. To the best of our knowledge, this is the first report to demonstrate the impact of preoperative OVF on prognosis for malignancies.

Vertebral fractures limit activity due to pain and loss of independence [11] and can be more associated with negative impacts on quality of life, the activities of daily living, and mortality than other patient-related factors. The mechanisms of the significant impact of OVF on poor prognosis have not been elucidated. Vertebral fracture is a common consequence of osteopenia or osteoporosis which is characterized by a low BMD [11]. In the present study, 69% of patients with OVF had osteopenia, and osteopenia was an independent risk factor for OVF. Although it is also not clear whether a low BMD is caused by cancer development or whether a low BMD promotes cancer development, there have been several possible molecular mechanisms presented regarding the relationship between BMD and cancer development.

There are direct and indirect relationships between malignancies and bone metabolism [18]. Bone loss is mainly caused by the activation of osteoclastogenesis via the RANK/RANKL (receptor activator of nuclear factor kappa-B ligand) signaling in response to proinflammatory and proosteolytic cytokines such as interleukin-1, 6, and tumor necrosis factor-α secreted by cancer cells [19]. The RANK/RANKL system is associated with bone metabolism and cancer development because the cancer cell microenvironment expresses RANK/RANKL and contributes to cancer development [20]. The RANKL released by osteoblasts stimulates the expression of RANK on the surface of the osteoclasts and enhances osteoclastogenesis. And during bone resorption, various cytokines such as transforming growth factor β and insulin-like growth factor that promote cancer growth, invasion, and metastasis are released [21]. These findings suggest that the RANK/RANKL system may be involved in both bone metabolism and cancer development.

In terms of the long-term outcomes of patients with OVF, patients with incident vertebral fractures were reported to have an increased risk of mortality (hazard ratio = 1.32) due to weight loss and physical frailty [22]. Six patients with OVF (18%) died of non-colorectal-cancer-related deaths (pneumonia in two patients, arrhythmia in one patient, and other causes in three patients), which could suggest that the presence of OVF might reduce the long-term OS due to decreased activities of daily living.

Moreover, osteopenia and osteoporosis are often associated with the long-term complications of various cancer treatments, including chemotherapy, radiotherapy, and hormone therapy, defined as cancer-treatment-induced bone loss (CTIBL) [23]. Our study demonstrated adjuvant chemotherapy for primary colorectal cancer as an independent risk factor for OVF. Oxaliplatin, included in one of the main adjuvant regimens for colorectal cancer, is a platinum compound. Oxaliplatin induces kidney failure caused by proximal tubule injury, resulting in hypomagnesaemia, which aggravates bone loss by preventing vitamin D synthesis [24]. In addition, vitamin D has been reported to suppress the growth of malignancies including colorectal cancer [25,26]. Vitamin D deficiency, which reduces BMD, may be one of the causes of poor prognosis in patients with OVF.

Interestingly, in this study, according to the Kaplan–Meier curve separated by bone status, the survival of patients with OVF was significantly lower than that of patients without OVF regardless of osteopenia. Osteoporosis is diagnosed using BMD and the presence of fracture [27]. Patients with vertebral fracture can be diagnosed with osteoporosis regardless of BMD, which suggests that patients with osteoporosis might have a worse prognosis than patients without osteoporosis. However, BMD measurement using DEXA is a widely recognized diagnosis tool for osteoporosis in the orthopedics area. We consider that evaluation of OVF using CT could be a substitute for the diagnosis of osteoporosis because DEXA remains an uncommon tool in the hepato-biliary-pancreatic surgery area.

Recurrence is not a strong prognostic factor for patients with resectable CRLM because repeat resection of recurrence also provides a survival benefit [28]. In this study, there was no difference in selecting the treatment of recurrence between patients with and without OVF. Notably, the positive impact of repeat resection for recurrence on cancer-specific survival rates was observed only in the non-OVF group (Appendix A), which suggested that OVF is useful not only for the preoperative stratification of prognostic factors but also for selecting the treatment of recurrence.

The present study includes several limitations. It is a retrospective analysis from a single institution with a small and homogeneous cohort. Therefore, the significance of OVF should be validated using other cohorts. In this study, OVF was identified using quantitative measurements and preoperative sagittal CT image reconstruction from the 11th thoracic vertebra to the 5th lumber vertebra. The current assessment for vertebral fracture is generally performed using spinal radiographs with a semiquantitative approach [29]. Therefore, further investigation is needed to determine the appropriate evaluation of OVF.

Preoperative treatment for OVF is difficult because most patients with OVF are asymptomatic. However, patients with OVF are elderly, malnourished, and have a low bone mineral density, and OVF also affects the treatment outcomes in the event of recurrence. Therefore, we believe that postoperative exercise, nutritional therapy, and osteoporotic drugs such as vitamin D, calcium, teriparatide, denosumab, and bisphosphates will contribute to improving the quality of life and prognosis after surgery. Moreover, the evaluation of BMD before adjuvant chemotherapy for primary colorectal cancer and the above interventions for osteoporosis to prevent OVF during chemotherapy are expected to improve the outcomes after hepatic resection for CRLM.

## 5. Conclusions

We demonstrated that preoperative OVF evaluated using CT was significantly associated with both a worse DFS and OS in patients who underwent hepatectomy for CRLM. The evaluation of preoperative OVF might be a useful prognostic indicator for patients with CRLM.

## Figures and Tables

**Figure 1 cancers-15-05513-f001:**
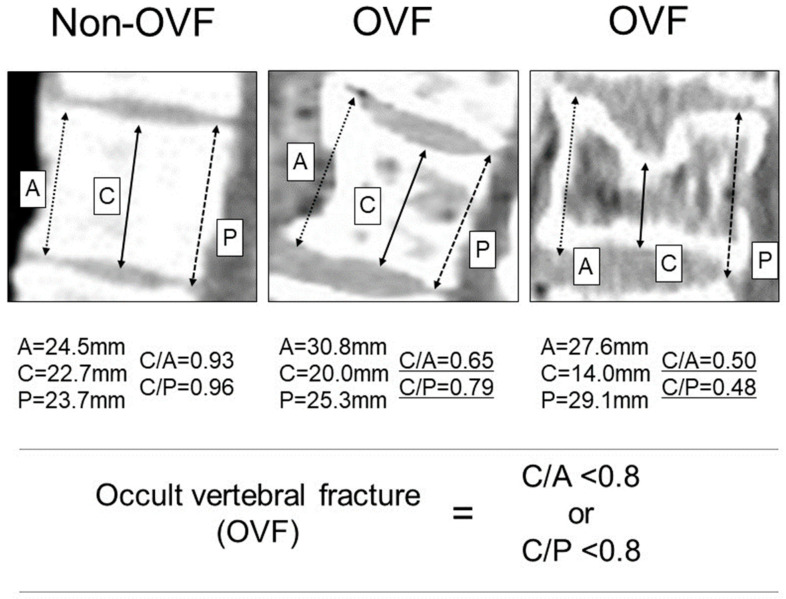
Identification of occult vertebral fracture (OVF) using a semiquantitative method and preoperative sagittal computed tomography image reconstruction. Anterior (A), central (C), and posterior (P) heights.

**Figure 2 cancers-15-05513-f002:**
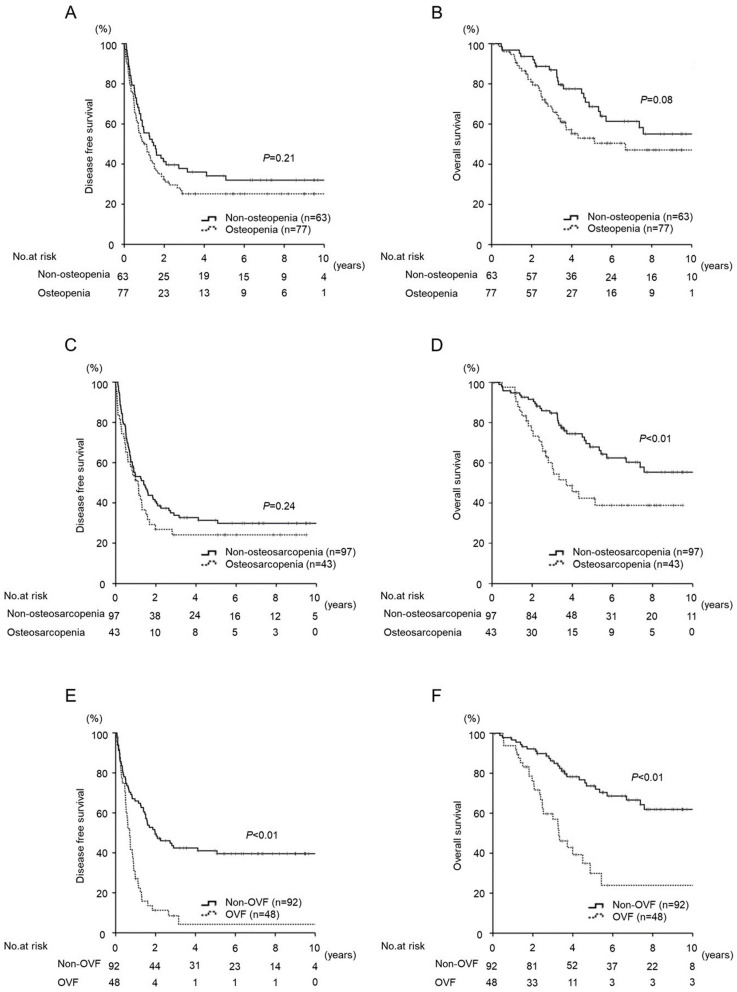
Kaplan–Meier curve for disease-free and overall survival after hepatic resection for colorectal liver metastases according to osteopenia, osteosarcopenia, and occult vertebral fracture (OVF). (**A**,**B**) Osteopenia, (**C**,**D**) Osteosarcopenia, (**E**,**F**) OVF.

**Figure 3 cancers-15-05513-f003:**
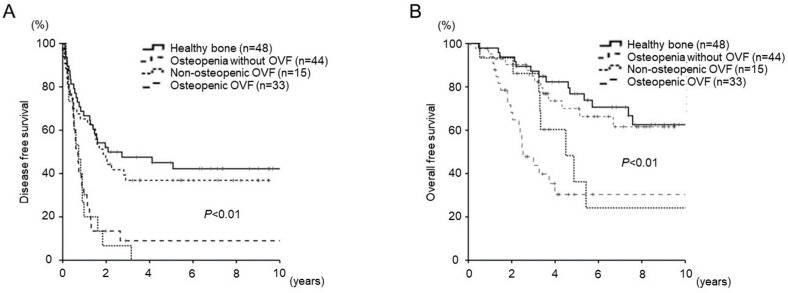
Kaplan–Meier curve for (**A**) disease-free survival and (**B**) overall survival after hepatic resection for colorectal liver metastases according to the bone status.

**Figure 4 cancers-15-05513-f004:**
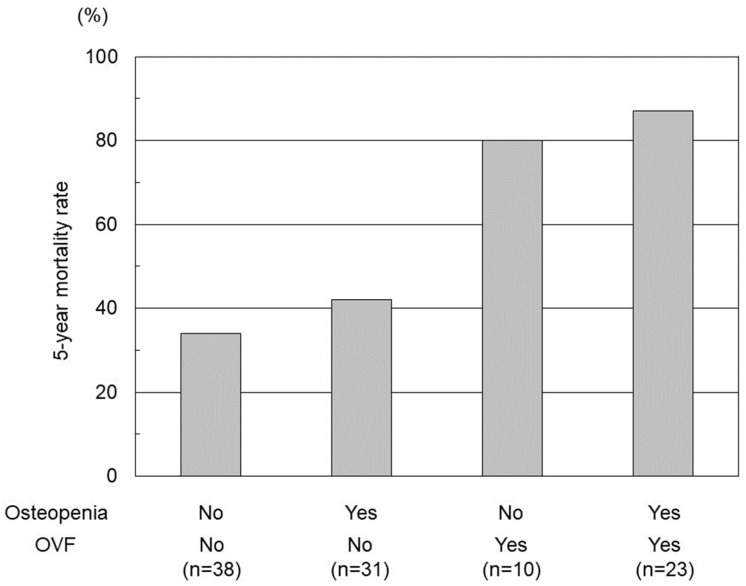
The rate of 5-year overall survival with or without osteopenia and/or occult vertebral fracture (OVF).

**Figure 5 cancers-15-05513-f005:**
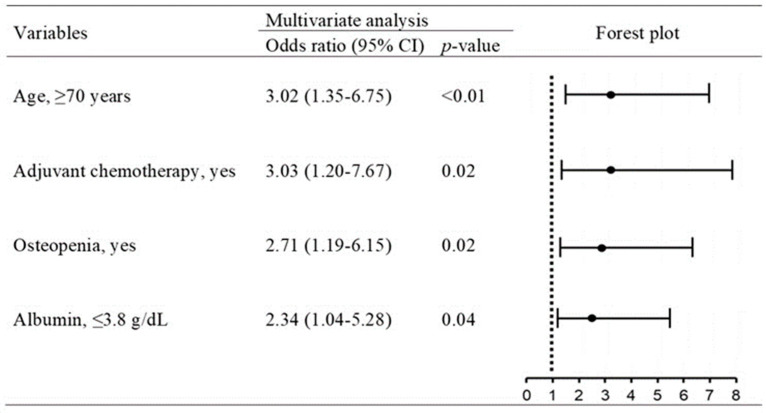
Multivariate logistic regression analysis for OVF.

**Table 1 cancers-15-05513-t001:** Clinicopathological variables associated with disease-free survival after hepatectomy for colorectal liver metastases using univariate and multivariate analyses.

Variables	N	DFS Univariate Analysis		DFS Multivariate Analysis	
		Hazard Ratio (95% CI)	*p*-Value	Hazard Ratio (95% CI)	*p*-Value
Lymph node metastases					
Yes	90	2.11	<0.01	1.89	<0.01
No	50	(1.35–3.31)		(1.18–3.01)	
Timing of tumor					
Synchronous	86	1.10	0.91		
Metachronous	54	(0.89–1.35)			
Resectability					
Resectable	76	0.73	0.12		
Borderline resectable or unresectable	64	(0.49–1.09)			
Neoadjuvant chemotherapy					
Yes	48	1.02	0.83		
No	92	(0.83–1.26)			
Tumor number					
Multiple	74	1.31	<0.01	1.28	0.02
Solitary	66	(1.07–1.61)		(1.03–1.57)	
Tumor size, mm					
>50	26	1.04	0.96		
≤50	114	(0.81–1.34)			
Extrahepatic lesion					
Yes	22	1.53	<0.01	1.74	<0.01
No	118	(1.19–1.96)		(1.34–2.26)	
Osteopenia					
Yes	77	1.14	0.21		
No	63	(0.93–1.39)			
Sarcopenia					
Yes	69	1.11	0.29		
No	71	(0.91–1.36)			
Osteosarcopenia					
Yes	43	1.29	0.24		
No	97	(0.84–1.97)			
OVF					
Yes	48	1.67	<0.01	1.70	<0.01
No	92	(1.36–2.06)		(1.38–2.11)	
GPS					
1 or 2	32	1.23	0.07		
0	108	(0.98–1.54)			
PNI					
≥45	84	0.89	0.24		
<45	56	(0.73–1.08)			
Serum CEA, ng/mL					
≥20	49	1.12	0.28		
<20	91	(0.91–1.37)			
Operative time, min					
≥420	55	1.21	0.06		
<420	85	(0.99–1.47)			
Intraoperative bleeding, g					
≥450	71	1.22	0.047	1.25	0.04
<450	69	(1.00–1.49)		(1.02–1.55)	
Infectious postoperative complication					
Yes	15	1.26	0.12		
No	125	(0.94–1.69)			
Curability					
R1 or 2	19	1.57	<0.01	1.51	<0.01
R0	121	(1.22–2.02)		(1.16–1.96)	

CEA: carcinoembryonic antigen, CI: confidence interval, DFS: disease-free survival, GPS: Glasgow Prognostic Score, OVF; occult vertebral fracture, PNI: prognostic nutrition index.

**Table 2 cancers-15-05513-t002:** Clinicopathological variables associated with overall survival after hepatectomy for colorectal liver metastases using univariate and multivariate analyses.

Variables	N	OS Univariate Analysis		OS Multivariate Analysis	
		Hazard Ratio (95% CI)	*p*-Value	Hazard Ratio (95% CI)	*p*-Value
Lymph node metastases					
Yes	90	1.90	0.04	1.18	0.30
No	50	(1.04–3.50)		(0.86–1.63)	
Timing of tumor					
Synchronous	86	1.24	0.45		
Metachronous	54	(0.71–2.18)			
Resectability					
Resectable	76	0.62	0.08		
Borderline resectable or unresectable	64	(0.36–1.05)			
Neoadjuvant chemotherapy					
Yes	48	1.24	0.44		
No	92	(0.72–2.15)			
Tumor number					
Multiple	74	1.87	0.03	1.21	0.21
Solitary	66	(1.08–3.25)		(0.90–1.62)	
Tumor size, mm					
>50	26	1.32	0.08		
≤50	114	(0.97–1.67)			
Extrahepatic lesions					
Yes	22	1.52	<0.01	1.79	<0.01
No	118	(1.11–2.08)		(1.28–2.51)	
Osteopenia					
Yes	77	1.28	0.08		
No	63	(0.97–1.67)			
Sarcopenia					
Yes	69	1.36	0.03	0.98	0.91
No	71	(1.04–1.79)		(0.66–1.45)	
Osteosarcopenia					
Yes	43	2.08	<0.01	2.54	0.02
No	97	(1.21–3.55)		(1.15–5.57)	
OVF					
Yes	48	1.85	<0.01	1.92	<0.01
No	92	(1.41–2.43)		(1.43–2.58)	
GPS					
1 or 2	32	1.39	0.03	1.05	0.78
0	108	(1.03–1.86)		(0.73–1.53)	
PNI					
≥45	84	0.75	0.03	0.87	0.40
<45	56	(0.57–0.97)		(0.62–1.21)	
Serum CEA, ng/mL					
≥20	49	1.27	0.08		
<20	91	(0.97–1.65)			
Operative time, min					
≥420	55	1.43	<0.01	1.33	0.06
<420	85	(1.10–1.87)		(0.99–1.80)	
Intraoperative bleeding, g					
≥450	71	1.24	0.11		
<450	69	(0.95–1.63)			
Infectious postoperative complication					
Yes	15	1.57	0.01	1.46	0.06
No	125	(1.10–2.26)		(0.99–2.16)	
Curability					
R1 or 2	19	1.33	0.14		
R0	121	(0.91–1.94)			

CEA: carcinoembryonic antigen, CI: confidence interval, GPS: Glasgow Prognostic Score, OS: overall survival, OVF; occult vertebral fracture, PNI: prognostic nutrition index.

**Table 3 cancers-15-05513-t003:** Univariate analysis of clinical variables in relation to OVF.

Variables	OVF		*p*-Value
	Yes (n = 48)	No (n = 92)	
Age, years	71 (62–76)	65 (58–71)	0.02
Gender, female	13 (27%)	31 (34%)	0.42
Body mass index, kg/m^2^	22.0 (19.3–23.9)	22.4 (20.6–24.4)	0.16
Lymph node metastases, yes	34 (71%)	56 (61%)	0.24
Adjuvant chemotherapy for a primary lesion before metachronous liver metastases, yes	15 (31%)	12 (13%)	0.01
Neoadjuvant chemotherapy, yes	14 (29%)	34 (37%)	0.36
Extrahepatic lesion, yes	5 (10%)	17 (19%)	0.21
Tumor number	2 (1–2)	2 (1–3)	0.87
Tumor size, mm	23 (15–37)	27 (17–45)	0.15
Serum CEA, ng/mL	17 (4–48)	8 (4–29)	0.15
Osteopenia, yes	33 (69%)	44 (48%)	0.02
Sarcopenia, yes	25 (52%)	44 (48%)	0.63
Osteosarcopenia, yes	30 (63%)	23 (25%)	0.12
Albumin, g/dL	3.7 (3.5–4.0)	3.9 (3.6–4.2)	0.01
Operation time, min	364 (239–495)	374 (288–470)	0.86
Intraoperative bleeding, g	390 (128–838)	450 (143–1088)	0.57
Infectious postoperative complicationa, yes	6 (13%)	9 (10%)	0.62
Postoperative stay, days	12 (9–14)	12 (9–17)	0.99
Curability, R1 or 2	7 (15%)	12 (13%)	0.80
Treatment of recurrence (resection:chemotherapy:others)	15:21:7	24:20:11	0.46

CEA: carcinoembryonic antigen, GPS: Glasgow Prognostic Score, OVF; occult vertebral fracture, PNI: prognostic nutrition index.

## Data Availability

The data from this study are available from the corresponding author [K.F.] on request.

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
