# Peer review of "Occult Vertebral Fracture (OVF) in Patients Who Underwent Hepatectomy for Colorectal Liver Metastasis: Strong Association with Oncological Outcomes"

_cancers, 2023, doi:10.3390/cancers15235513_

Round 1
Reviewer 1 Report
Comments and Suggestions for Authors
This is an interesting paper regarding the occult vertebral fracture (OVF) on oncological results post-surgery for colorectal liver metastases (CRLM). 34% had OVF in this study’s cohort. For overall survival, extrahepatic lesions, osteosarcopenia, and OVF were independent and significant predictors. Age, prior adjuvant chemotherapy, osteopenia, and hypoalbuminemia increased OVF risk.
There are some comments to be addressed:
This study is an intriguing investigation that evaluated the preoperative OVF in patients who underwent hepatectomy for CRLM.
1. The intricate relationship between OVF and the postoperative prognosis of CRLM. While a variance in OS is logical, the factors influencing DFS remain opaque. Questions arise regarding whether OVF's presence sways the selection of treatment for recurrence, the frequency of repeat resection, and chemotherapy, and its impact on cancer-specific survival rates.
2. Absent a comprehensive understanding of the points above, delineating a clinical approach for patients diagnosed with preoperative OVF is challenging. Could enhancing nutritional intake and physical activity extend their prognosis? Or can OVF serve as an indicator of the biological aspect?
3. Tumor resectability and genetic mutations such as RAS/BRAF are consensus prognostic factors for CRLM. Does the study differentiate between resectable and borderline resectable CRLM? Given that roughly a third of the patients received neoadjuvant chemotherapy, clear criteria for neoadjuvant or adjuvant treatments are crucial. Incorporating your institution's treatment policy for CRLM in the methods section would add clarity.
4. How many individuals in your cohort underwent intensive CRLM treatments, such as conversion surgery for initially deemed unresectable cases? Prognostic disparities are expected between single or 2-3 CRLMs versus initially unresectable CRLM. The study's details on the 140-patient cohort are scant, rendering it challenging to ascertain if OVF evaluations are applicable across any CRLM cohort.
Comments on the Quality of English LanguageThere are generally no problems regarding the quality of the English language.
Author Response
Reviewer: 1
This is an interesting paper regarding the occult vertebral fracture (OVF) on oncological results post-surgery for colorectal liver metastases (CRLM). 34% had OVF in this study’s cohort. For overall survival, extrahepatic lesions, osteosarcopenia, and OVF were independent and significant predictors. Age, prior adjuvant chemotherapy, osteopenia, and hypoalbuminemia increased OVF risk.
There are some comments to be addressed:
This study is an intriguing investigation that evaluated the preoperative OVF in patients who underwent hepatectomy for CRLM.
- The intricate relationship between OVF and the postoperative prognosis of CRLM. While a variance in OS is logical, the factors influencing DFS remain opaque. Questions arise regarding whether OVF's presence sways the selection of treatment for recurrence, the frequency of repeat resection, and chemotherapy, and its impact on cancer-specific survival rates.
We appreciate your important comments. There was no difference in the frequency of repeat resection and chemotherapy for recurrence between patients with and without OVF (Table 3). On the other hand, the positive impact of repeat resection for recurrence on cancer-specific survival rates was observed only in non-OVF group (Supplementary Figure 1), which suggests that OVF is useful not only for preoperative stratification of prognostic factors but also for selecting treatment of recurrence. We described it in Discussion (l. 269-75). Thank you for your suggestion so much.
- Absent a comprehensive understanding of the points above, delineating a clinical approach for patients diagnosed with preoperative OVF is challenging. Could enhancing nutritional intake and physical activity extend their prognosis? Or can OVF serve as an indicator of the biological aspect?
Patients with OVF were elderly, malnourished, and have low bone mineral density as the biological aspect. And as you suggested, OVF also affects recurrence treatment outcomes. Therefore, we believe that postoperative nutritional intake including osteoporotic drugs and physical activity contribute to improve not only their quality of life but also prognosis. We described it in Discussion (l. 285-9).
- Tumor resectability and genetic mutations such as RAS/BRAF are consensus prognostic factors for CRLM. Does the study differentiate between resectable and borderline resectable CRLM? Given that roughly a third of the patients received neoadjuvant chemotherapy, clear criteria for neoadjuvant or adjuvant treatments are crucial. Incorporating your institution's treatment policy for CRLM in the methods section would add clarity.
Thank you for your comment. We added resectability as a variable (Table 1 and 2). Unfortunately, genetic mutation information could not be added as a variable due to many missing data in our database. Also, we described our institution's treatment policy for CRLM including the definition of resectability and chemotherapy indication in the methods section (l.72-9).
- How many individuals in your cohort underwent intensive CRLM treatments, such as conversion surgery for initially deemed unresectable cases? Prognostic disparities are expected between single or 2-3 CRLMs versus initially unresectable CRLM. The study's details on the 140-patient cohort are scant, rendering it challenging to ascertain if OVF evaluations are applicable across any CRLM cohort.
We appreciate youe suggestion. This cohort included only 3 patients who underwent intensive conversion hepatectomy for initially unresectable CRLM. No significant difference in prognosis was observed in resectability (Table 1 and 2). And as you suggested, we added detailed information of this cohort in Patients’ characteristics session (l. 126-30).

Reviewer 2 Report
Comments and Suggestions for Authors
This is a retrospective study on the impact of occult vertebral fracture (OVF) on the oncological outcome after hepatectomy for CLM. The study is interesting and of clinical relevance since the detection of OVF before surgery has a negative impact on patients' long-term survival. I have some concerns as listed below.
1. Need to explicit the hypothesis in the background of the abstract
2. OVF is somehow related to osteosarcopenia. However, only OVF is the significant factor for DFS but not osteosarcopenia. The authors need to provide reasons for that.
3. Further to point no.2, is there any correlation between OVF and osteosarcopenia?
4. Were those patients with hx of vertebral fracture, bone degeneration or congenital skeletal disorder excluded from the study, as these may affect the objective measurement of OVF ?
5. The description of patient selection, perioperative management, operative details and perioperative outcomes are all missing the manuscript.
6. The section on predictive factors for OVF is confusing. OVF is supposed to be independent variable accounting for poor prognosis. More details should be put on to explain how OVF can be related to poor oncological outcome.
Comments on the Quality of English LanguageSome minor editing on grammar is needed.
Author Response
Reviewer: 2
This is a retrospective study on the impact of occult vertebral fracture (OVF) on the oncological outcome after hepatectomy for CLM. The study is interesting and of clinical relevance since the detection of OVF before surgery has a negative impact on patients' long-term survival. I have some concerns as listed below.
- Need to explicit the hypothesis in the background of the abstract
Thank you for your kind suggestion so much. We added the hypothesis of this syudy in Abstract (l. 23-4)
- OVF is somehow related to osteosarcopenia. However, only OVF is the significant factor for DFS but not osteosarcopenia. The authors need to provide reasons for that.
- Further to point no.2, is there any correlation between OVF and osteosarcopenia?
We appereciate your constructive commets. As you suggested, both osetosarcopenia and OVF are frailty phenotypes, and we believe that they include a common group of patients. Patients with OVF had more osteosarcopenia than those without OVF (63% vs. 25%, p=0.12, Table 3). Although the discrepancy you pointed out have remaind unclear and need to be investigated in the future, we consider how many effective treatments for recurrence could be done might be related to osteosarcopeenia unlike OVF (Table 3 shows that there was no difference in the frequency of repeat resection and chemotherapy for recurrence between patients with and without OVF).
- Were those patients with hx of vertebral fracture, bone degeneration or congenital skeletal disorder excluded from the study, as these may affect the objective measurement of OVF ?
Thank you for your important comments. As you suggested, patiets with symptomatic vertebral fracture and other bone diseases were excluded in this study. We specified it in Methods (l. 71).
- The description of patient selection, perioperative management, operative details and perioperative outcomes are all missing the manuscript.
Thank you for your kind suggestion. As you suggested, we described the critetia of patients selection and operative management and cited references with detailed information (l. 67-81).
- The section on predictive factors for OVF is confusing. OVF is supposed to be independent variable accounting for poor prognosis. More details should be put on to explain how OVF can be related to poor oncological outcome.
We appreciate your suggestion. We added reasons why OVF can be related to poor long-term outcome in Discussion (l. 244-9, 257-9, and 269-75).

Round 2
Reviewer 1 Report
Comments and Suggestions for Authors
The author has responded appropriately to the comments and corrected the manuscripts. There are no additional comments.
Reviewer 2 Report
Comments and Suggestions for Authors
The authors have revised the manuscript sufficiently and can be accepted for publication.